# Data-Centric Diet: Effective Multi-center Dataset Pruning for Medical Image Segmentation

**Yongkang He** [1]  **Mingjin Chen** [1]  **Zhijing Yang** [1]  **Yongyi Lu** [1]

## Abstract

This paper seeks to address the dense labeling problems where a significant fraction of the dataset can be pruned without sacrificing much accuracy. We observe that, on standard medical image segmentation benchmarks, the loss gradient norm-based metrics of individual training examples applied in image classification fail to identify the important samples. To address this issue, we propose a data pruning method by taking into consideration the training dynamics on target regions using Dynamic Average Dice (DAD) score. To the best of our knowledge, we are among the first to address the data importance in dense labeling tasks in the field of medical image analysis, making the following contributions: (1) investigating the underlying causes with rigorous empirical analysis, and (2) determining effective data pruning approach in dense labeling problems. Our solution can be used as a strong yet simple baseline to select important examples for medical image segmentation with combined data sources.

## 1. Introduction

Training better deep neural networks often involve increasing the amount of training data and training over-parameterized models. But are all the samples necessarily needed for effective learning? Dataset pruning and training with smaller subsets (Coleman et al., 2019; Hwang et al., 2020) without performance degrading would reduce the memory and computation cost, potentially allowing for better deployment in resource-constrained environments such as mobile devices. However, finding important examples during training remains a challenging task.

While large numbers of previous works have explored dif-

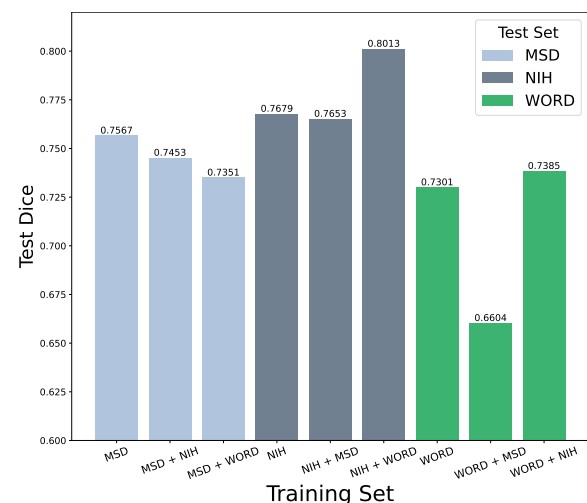

*Figure 1.* Results of a same 3D U-Net model trained with single or combined datasets. More data does not necessarily improve the segmentation performance.

ferent dataset pruning methods (Paul et al., 2021; Toneva et al., 2018; Feldman & Zhang, 2020; Agarwal et al., 2022; Sorscher et al., 2022; Killamsetty et al., 2021a; Mirza-soleiman et al., 2020; Killamsetty et al., 2021b) and investigated example difficulty (Baldock et al., 2021; Hacohen et al., 2020; Mangalam & Prabhu, 2019; Krymolowski, 2002) on classification tasks, in this paper, we take a step towards a better understanding of dense labeled data, and investigate the impact of dataset selection on medical image segmentation. The main questions we ask are: (i) Do dataset combinations necessarily lead to better segmentation performance? (ii) What types of data can be removed from the datasets without reducing the segmentation performance? (iii) How early in the training process can we identify those data? We start to answer these questions empirically from experimental perspectives in the context of popular medical image segmentation benchmarks.

Our first finding is that, combining multiple datasets for training does not necessarily yield better models for segmentation (see Fig.1. Take three popular pancreas segmentation datasets for example, combining WORD and MSD datasets for training surprisingly lowers the testing Dice

[1]Guangdong University of Technology, Guangdong, China. Correspondence to: Yongyi Lu <yylu@gdut.edu.cn>.

*Workshop on Interpretable ML in Healthcare at International Conference on Machine Learning (ICML)*, Honolulu, Hawaii, USA. 2023. Copyright 2023 by the author(s).

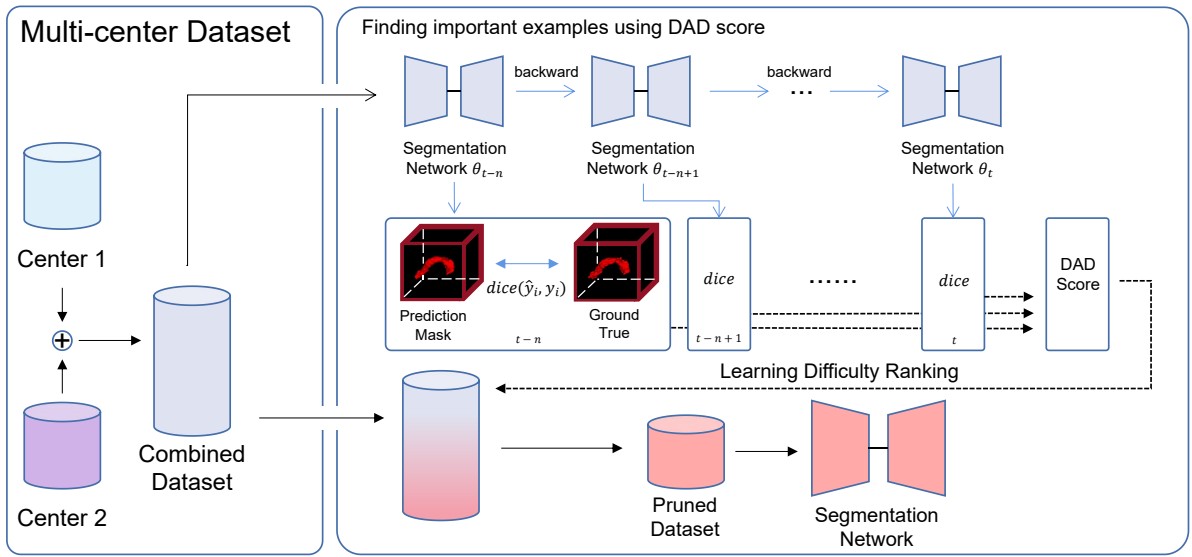

*Figure 2.* Our pipiline of using DAD score to prune dataset.

score compared to model trained solely on WORD. Similar performance degradation can be observed through MSD + NIH testing results. The increase of data variance may introduce a large amount of noise and redundancy, which is harmful to effective learning.

Here comes the second question: Can we identify the more-useful training examples and remove those less-useful or even harmful examples for better training? We make the striking observation that, on standard medical image segmentation benchmarks, the loss gradient norm-based metrics of individual training examples applied in image classification (such as GradNd (Paul et al., 2021), VoG (Agarwal et al., 2022)) fail to identify the important samples (see Fig.4 for a detailed comparison). These gradient-based pruning metrics often compute score of the whole image, which does not work on the dense labeling task, especially form small target prospective, where the computed scores are easily affected by the large background regions. This inspired us to re-design selected criteria for segmentation which focuses more on the target regions. To address this critical issue, we introduce Dynamic average Dice (DAD) score as an efficient and practical method to rank data according to the learning difficulty of target samples. Our framework of using DAD to pruning dataset is shown in Fig.2. Unlike previous methods, the proposed DAD score focuses more on the segmentation target itself than on the whole image, which is more effective for tasks with dense labels.

To better reveal the dynamics of the learning difficulty of the model for all samples during training, we we further design the moving distance of samples movement in the data map to determine if the rankings of samples have stabilized,

which naturally answers question (iii). The whole pipeline is rigorously validated in three medical image segmentation benchmarks as well as the combinations between them. For example, we can prune 40% of examples from NIH + MSD datasets with a 1% increase in Dice score surprisingly.

In summary, we make three primary contributions to answer the above questions:

- We systematically study different data sources and find that combining multiple sources does not necessarily yield better models for segmentation.

- We propose an efficient and practical method to rank data according to the learning difficulty of samples by computing the Dynamic average Dice (DAD) score.

- We rigorously validate its effectiveness for pruning datasets in popular medical image segmentation benchmarks and demonstrate that our method significantly outperforms the prior arts.

## 2. Related Work

Finding important training samples is a common thread of research into efficient deep learning. Online hard example mining were proposed to improve learning accuracy in object detection (Shrivastava et al., 2016) and classification (Chang et al., 2017). (Paul et al., 2021) introduced a 'Data Diet' to identify data that can be pruned in early training without sacrificing accuracy via individual initial loss gradient norms. (Toneva et al., 2018) developed a forgetting score which measures the degree of forgetting to prune datasets by only including frequently-forgotten examples.

(Feldman & Zhang, 2020) defined a memorization score for each example, with high memorization scores corresponding to hard examples that must be individually learned. (Agarwal et al., 2022) proposed variance of gradients as a valuable and efficient metric to rank data by difficulty. (Sorscher et al., 2022) developed a self-supervised pruning metric that demonstrates comparable performance to the best supervised metrics on ImageNet. Other investigations on example difficulty include (Baldock et al., 2021; Hacohen et al., 2020; Mangalam & Prabhu, 2019; Krymolowski, 2002). However, the performance of the above selection metrics on dense labeling tasks such as segmentation has not yet been explored. Recently, (Killamsetty et al., 2021a) proposed an adaptive data subset selection algorithm where subsets are chosen by minimizing the difference between the gradient of the subset and full data. They demonstrated that GRAD-MATCH achieves the best speedup accuracy tradeoff compared to prior arts (CRAIG (Mirzasoleiman et al., 2020), GLISTER (Killamsetty et al., 2021b)). But there is limited study discussing how to select the important examples when facing dense labeling tasks.

To summarize: there are easier and harder images, by identifying which can improve efficient learning. However, this relationship has not commonly been explored and recognised for segmentation task. Our study on medical image segmentation systematically presents the existence of the problem in dominant metrics used for image classification, and produces a comprehensive guideline of identifying important examples for image segmentation.

## 3. Methods

In this section, we present our framework to recognize learning difficulty of samples with dense labels, and according it to prune dataset. The pruned core-dataset has a smaller size and more efficient data distribution.

### 3.1. Example Learning Difficulty

Note Our goal is to rank examples according to learning difficulty of example in training process. We consider a supervised segmentation problem in medical image, a neural network is trained to segment a pancreas organ from CT image. Suppose a training set $S = (x_i, y_i)_{i=1}^N$ from an unknown data distribution $D$, where $x_i$ denotes a CT image of size $W \times H \times L$, and its segmentation map of organ $\hat{y}_i$ generated by neural network.

Similar to how school teachers use student scores to assess the difficulty of a particular question, an intuitive way to measure the learning difficulty of a sample is to directly count its average score during training:

$$\hat{\mu}_i(T) = \frac{1}{T} \sum_{e=0}^{T} \|x_i - y_i\|_2 \tag{1}$$

This is also usually a common measure of learning difficulty in active and continuous learning. However, we note that this calculation suffers from category imbalances in segmentation tasks, such as the pancreas, which represents only a small fraction of the volume of the entire abdomen CT image so the score will be dominated by the background. Therefore, a better form is to use the dice score instead of the original equation:

$$\hat{\mu}_i(T) = \frac{1}{T} \sum_{e=0}^{T} dice(f(x_i; \theta_e), y_i) \tag{2}$$

where $\hat{y}_i = f(x_i; \theta_e)$ denotes the segmentation map predicted by model with parameters $\theta_e$ at each epoch $e$.

Based on Eq.2, we define **DAD** (Dynamic average dice) score to rank all examples globally. For a specific training epoch $e = t$, calculate the average dice coefficient of sample $x_i$:

$$\hat{\mu}_i(t) = \frac{1}{\Delta t} \sum_{e=t-\Delta t}^{t} dice(f(x_i; \theta_e), y_i) \tag{3}$$

We use the sample's DAD score over time period t as its dynamic learning difficulty, and recommend a value of 10 or greater for $\Delta t$. A obvious question is, why use the dynamic average of dice scores $\hat{\mu}_i(t)$ rather than $\hat{\mu}_i(T)$?

### 3.2. An Analytic Theory of DAD Score

Following several seminal papers in explainability literature, there are distinct stages to training in deep neural networks(Jiang et al., 2020; Mangalam & Prabhu, 2019; Faghri et al., 2020; Achille et al., 2017). Then Eq.2 can be decomposed into:

$$\hat{\mu}_i(T) = \frac{1}{T} \left( \sum_{e=0}^{t_1} d_{i,\theta_e} + \sum_{e=t_1}^{t_2} d_{i,\theta_e} + \sum_{e=t_2}^{T} d_{i,\theta_e} \right) \tag{4}$$

where we consider three training period $e \in [1, t_1], e \in [t_1, t_2]$ and $e \in [t_2, T]$, i.e. time after the initialization of the early network, the period of time when the loss value decreases rapidly, and the period of time when the network gradually converges to over-fitting, respectively. For this formula, consider two cases:

#### 3.2.1. LATE PRUNING

***Use long time to train the segmentation model to convergence, i.e.*** $T \to \infty$, $\hat{\mu}_i(T)$ will gradually approach to a

Easy-to-learn  Hard-to-learn

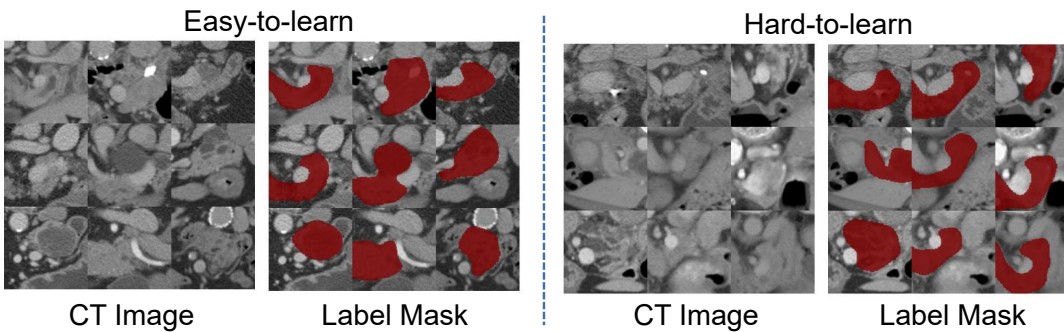

CT Image    Label Mask      CT Image    Label Mask

*Figure 3.* The 9 easiest and hardest images to learn in the MSD dataset, it is clearly observed that the images of the hard-to-learn example are more blurred and the pancreas is easily confused with other organs.

constant:

$$\lim_{T \to \infty} \hat{\mu}_i(T) = \lim_{T \to \infty} \frac{1}{T}(t_1 \xi_{t_1} + (t_2 - t_1)\xi_{t_2} + (T - t_2)\xi_T)$$
$$= \xi_T \approx 1$$
(5)

where $\xi_t, t \in [t_1, t_2, T]$ denotes the mean dice of sample $x_i$ in three training period $e \in [1, t_1], e \in [t_1, t_2]$ and $e \in [t_2, T]$. In this case, the dice scores of all samples will converge to 1 due to the over-fitting of the neural network, so cannot to distinguish the learning difficulty of the samples by $\hat{\mu}_i$.

### 3.2.2. EARLY PRUNING

***When calculate it in early training i.e.*** $T \to t_1$**:** in this case, Eq.4 will trans to $\lim_{T \to t_1} \hat{\mu}_i(T) = \xi_{t_1}$, which contain information about errors, because network gradient norm at initialization will be influenced by the random parameterization method, in other words, $\xi_{t_1}$ contains different noise to a large extent.

In summary, in order to accurately calculate the example learning difficulty, it is better to calculate the dice score for each sample for a period of time before the network converges, *i.e.*, $\hat{\mu}_i(t), t \in (t_1, t_2)$.

### 3.3. DAD Score with Moving Distance

In previous section, we analyse why use the dynamic average of dice scores $\hat{\mu}_i(t)$ rather than $\hat{\mu}_i(T)$. The following question is, how to choose the right time $t$ to calculate DAD?

To measure whether the model's learning of a sample is stable, we follow (Swayamdipta et al., 2020) to define variability of our DAD, which indicates whether the sample is easy to be forgotten by the model:

$$\hat{\sigma}_i = \sqrt{\frac{\sum_{e=t_0-t}^{t_0}(dice(f(x_i; \theta_e), y_i) - \hat{\mu}_i)^2}{t}}$$
(6)

We use the moving distance curve $L$ to measure the change

of examples learning difficulty:

$$L = \sum_{i=1}^{n}(\Delta\mu_i + \Delta\sigma_i)$$
(7)

where $\Delta\mu_i = \mu_i^{t_0} - \mu_i^{t_0-t}$ and $\Delta\sigma_i = \sigma_i^{t_0} - \sigma_i^{t_0-t}$ represents the difference between the DAD score and its variability calculated each time and its value calculated at the previous time, and $n$ denotes the number of samples in the training set. According to Eq.7, we can know exactly when we can finish training in advance and find important examples.

As shown in Fig.6(b), the moving distance curve $L$ accurately reflects when the DAD ranking of the training set stabilizes. We assume that we stop training at 1% of the maximum value of the moving distance curve and prune the dataset at this point. Our results demonstrate that the subset selected by this method does not differ significantly from the one reselected later, and the final test results (blue line) are very similar. This indicates that our method is capable of accurately determining the optimal time for dataset pruning, providing a method to speed up training.

### 3.4. Overall Pipeline

Our goal is to develop a framework that find a subset of data points that maintain a similar level of quality. The segmentation Network training on the subset yields comparable or better performance than training on the entire dataset. Our framework comprises two stages as shown in Fig.2. In the first stage, a segmentation network is trained on the complete combined dataset, and the DAD score of each sample is calculated during training. Algorithm 1 recaps the dataset pruning process. In the second stage, the most important data are selected to form a new subset based on the ranking of samples according to their DAD scores. A new segmentation network is then trained on the pruned subset. The algorithm flow is shown in Algorithm.1.

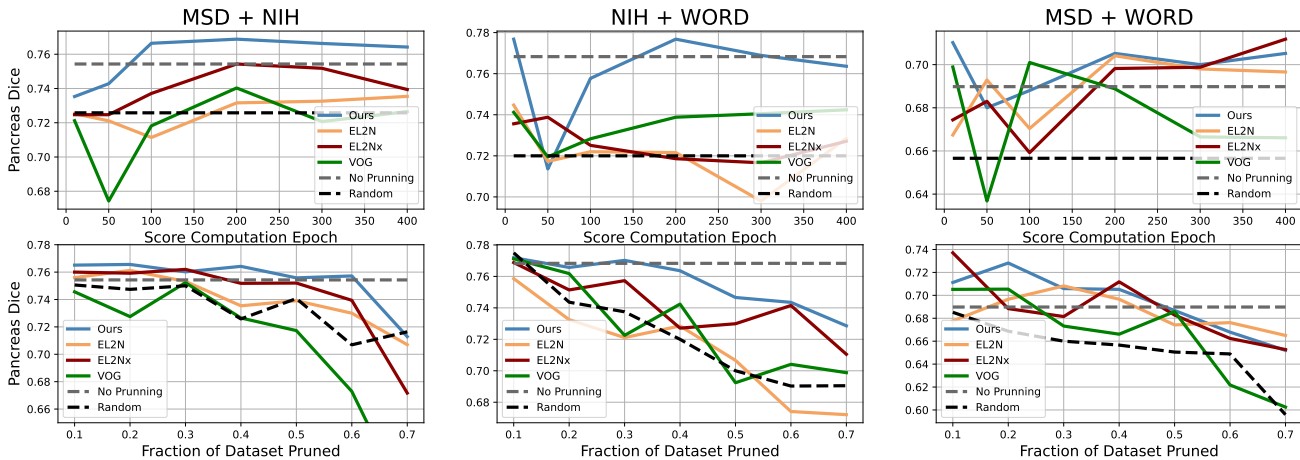

*Figure 4.* Each column represents a combination of two different data sets. First row: Each score is calculated at different periods of training, and a subset of 60% size of the complete dataset (i.e., pruning ratio of 0.4) is selected for retraining after ranking the samples. Second row: The final accuracy obtained by training after pruning the dataset by different fractions. For comparison purposes, the individual scores at the 400-th epoch are selected uniformly for pruning.

# 4. Applications

## 4.1. Data Pruning

For the purpose of enhancing model generalization performance, researchers may use a combined dataset from multiple sources for training. However, without data selection, domain gap may hinder the training of the model in terms of performance degradation. Therefore data pruning on a collection of multiple datasets is a matter of concern. Our algorithm of data pruning is shown in Algorithm.1.

---

**Algorithm 1** Dataset Pruning Process

---

**Input:**
$D \in \{x_i\}$      # Dataset to be pruned
$F(.)$      # Segmentation Model
$t$      # DAD score calculation interval
$p$      # Dataset pruning rate
$L_{max} \leftarrow 0$

**for** $e = 1$ to $T$ **do**
    $d_{i,e} \leftarrow Dice(F(x_i), y_i), \forall i \in [n]$
    $t_0 \leftarrow e$
    $\hat{\mu}_i \leftarrow mean(d_{i,e}), e \in [t_0 - t, t_0]$    # Eq.3
    $\hat{\sigma}_i \leftarrow var(d_{i,e}), e \in [t_0 - t, t_0]$    # Eq.6
    $L_{max} \leftarrow max(L_{max}, L(\Delta\hat{\mu}_i, \Delta\hat{\sigma}_i))$    # Eq.7
    **if** $L(\Delta\hat{\mu}_i, \Delta\hat{\sigma}_i) < 0.01 L_{max}$ **:**
        **break**
    **backward** $F(.)$
**end for**
$R = sort(\hat{\mu}_i)$      # DAD Score Ranking
$D_{pruned} \leftarrow$      # Pruned Dataset
    $D(R[0.5pn : 0.5(1-p)n])$

---

### 4.1.1. DATA PRUNING IN DIFFERENT FRACTION

In Fig.4, we show the final results of various methods for pruning on the three different combined dataset. More details about other methods and implementation see Sec.5.2. Our method performs almost best at different pruning scales. Moreover, experiment results suggest that the model trained on the pruned dataset showed better performance than that trained on the full combined dataset, demonstrating that a larger dataset size is not always better.

In particular, we observe that our proposed EL2Nx has better performance than EL2N. The EL2N score measures the error between the model's prediction for the entire image and the ground truth, while EL2Nx only considers the error for the target organ. However, in CT images, the pancreas usually occupies only a small fraction of the total area, causing the EL2N score to be overwhelmed by background pixels. This issue also affects the VOG scores, resulting in poor performance for both metrics on the segmentation task. In contrast, the DAD score addresses the issue by not only considering the error of the organ itself, but also incorporating artifacts and noise in the model output. This provides a more accurate reflection of the learning difficulty of the sample, making it a useful metric for evaluating the performance of segmentation models.

Nevertheless, we found that a sharp drop in performence when pruning early in training, as shown in the first column of Fig.4. We hypothesize that this is because the score at the beginning of training does not reflect the true learning difficulty of the samples. With a specific metric, it is likely that stable score rankings can be obtained in the middle of training, without the need for complete training until the model converges. We verify the assumption in the next

section and give a feasible solution.

### 4.2. Utility of DAD as an Auditing Tool

**Qualitative inspection of ranking:** Qualitative inspection of examples with high and low DAD scores showed that images at either end of the rankings had different semantic properties. We visualize 18 images ranked lowest and highest according to DAD for both the entire dataset for MSD in Fig.3. It shows that CT images of hard-to-learn exhibit a high degree of similarity: they both have low contrast and blurred backgrounds, while the boundaries between organs are fuzzy and difficult to distinguish. While images of easy-to-learn are much clearer and has a higher contrast, while the organ shapes are also similar.

**Training dynamic visualizations:** DAD score can use to be DataMap(Swayamdipta et al., 2020) for visualize the network training process. We visualize the dynamic change process of data maps during the training process for different dataset combinations as shown in Fig.5.

## 5. Experiments

### 5.1. Datasets

We used three public CT datasets: MSD(Simpson et al., 2019)(100 cases), NIH(80 cases) and WORD(100 cases) for pancreas segmentation, each dataset is randomly spilt into 80% of training data and 20% of testing data.

**MSD-Pancreas** [1] contains 420 portal-venous phase 3D CT scans (282 Training and 139 Testing), having labels of pancreas and tumor. The CT scans have resolutions of 512 × 512 × $l$ pixels. We merge the pancreas and tumor labels together as pancreas in our task. To ensure that the amount of data from different centers is roughly balanced, we randomly selected 100 cases from MSD dataset, and then randomly divided them into 80 training cases and 20 test cases.

**WORD** [2] contains 150 abdomen CT scans. Each CT volume consists of 159 to 330 slices of 512 × 512 pixels, with an in-plane resolution of 0.976 mm × 0.976 mm and slice spacing of 2.5 mm to 3.0 mm, acquired on a Siemens CT scanner. We randomly selected 100 cases from WORD dataset, and then randomly divided them into 80 training cases and 20 test cases.

**NIH-Pancreas** [3] Pancreas CT contains 82 abdominal contrast enhanced 3D CT scans. The CT scans have resolutions of 512 × 512 pixels with varying pixel sizes and slice

---

[1]http://medicaldecathlon.com/

[2]https://github.com/HiLab-git/WORD

[3]https://wiki.cancerimagingarchive.net/display/Public/Pancreas-CT

thickness between $1.5 \sim 2.5$mm, acquired on Philips and Siemens MDCT scanners. The dataset is randomly splitted into a training set of 60 training cases and 22 test cases.

### 5.2. Experimental Setup

#### 5.2.1. IMPLEMENTATION DETAILS

We use 3D U-Net as our backbone segmentation network, which is optimized by Adam solver for 25000 iterations with the batch size of 4 and a initial learning rate of 0.003. Each patch has size $64 \times 64 \times 64$. For data pruning experiment, we first run a baseline on the entire dataset and then prune the training set using the DAD score to obtain a subset. Then the segmentation network is initialized and retrained on the new subset. All experiments are run on a 24GB NVIDIA GeForce RTX 3090, used about 4000 GPU hours for entire experiments.

#### 5.2.2. COMPARISON WITH OTHER METHODS.

In Sec 4.1.1, we compare the effects of different metrics (random, VOG(Agarwal et al., 2022), EL2N(Paul et al., 2021), EL2Nx and ours method) used to prune datasets.

- The VOG (Variance of Gradients) score is a class-normalized gradient variance score for determining the relative ease of learning data samples within a given class.

- The EL2N of a training sample x is defined to be $E \left\| p(w_t, x) - y \right\|_2$, where $p(w_t, x)$ denote the neural network output after softmax function in the form of a probability vector.

- We improved EL2N to validate the importance of focusing on the learning difficulty of the foreground, which is EL2Nx. It defined to be $E \left\| p(w_t, x_f) - y_f \right\|_2$, it only calculates the 2-norm of the error between the image and label on the foreground target i.e. $x_f$ and $y_f$.

### 5.3. Ablation Experiment

#### 5.3.1. INSTANCES WITH DIFFERENT DIFFICULTY

Fig.3 presents the instances have the high DAD score. For these samples, the model always has good predictive power during training; thus, we refer to them as easy-to-learn. In contrast, we call the samples that have low DAD score as hard-to-learn. Finally, we refer to the intermediate data beyond these two parts as ambiguous data.

Many past studies have reported that it is better to keep hard-to-learn examples than easy-to-learn examples because the latter are usually more numerous, creating redundancy in the data (Agarwal et al., 2022; Toneva et al., 2018). But these

Data Maps of (MSD+WORD)

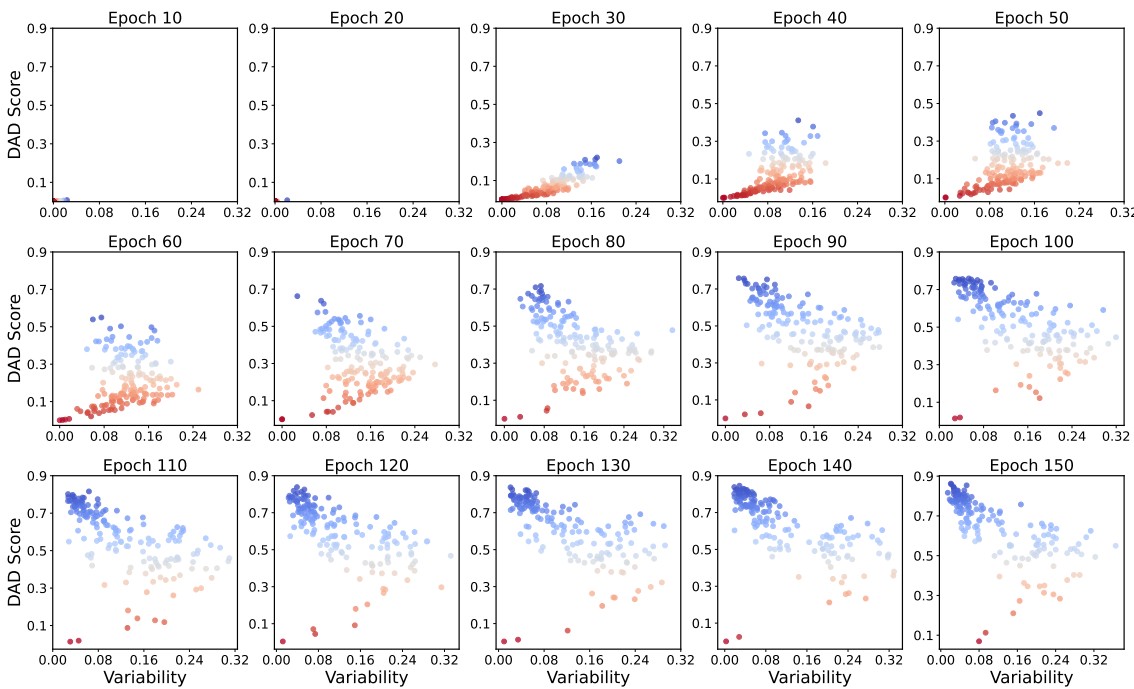

*Figure 5.* Data Maps for combined dataset of MSD and WORD.

|  | MSD | | | NIH | | | WORD | | |
|---|---|---|---|---|---|---|---|---|---|
| subset size | 100% | 66% | 33% | 100% | 66% | 33% | 100% | 66% | 33% |
| random | | 0.7258 | 0.5018 | | 0.7496 | 0.6775 | | 0.7140 | 0.6576 |
| hard-to-learn | 0.7567 | 0.7042 | 0.5338 | 0.7679 | 0.7282 | 0.7008 | 0.7301 | 0.6872 | 0.5840 |
| easy-to-learn | | 0.7501 | 0.6222 | | 0.7514 | 0.7086 | | 0.7294 | 0.6399 |
| ambiguous | | **0.7642** | **0.6414** | | **0.7568** | **0.7138** | | **0.7316** | **0.6983** |

*Table 1.* Comparison of DSC score for V-Net models trained on different regions selected by DAD score. The results show that models trained on those ambiguous samples performed better.

studies were conducted on natural image datasets, which may not be applicable to medical images. On the other hand, recent work (Sorscher et al., 2022) has demonstrated that this theory is only applicable to large training sets, for small datasets, keeping the hardest examples performs worse than keeping others.

As shown on Table.1, our experiments show that preserving those ambiguous examples is better than preserving other regions, whether at low or high pruning ratios, and it presents consistency across datasets. Training ambiguous data is expected to enhance the performance of the model, even if the size of the data is reduced. Moreover, when model training on a subset of hard-to-learn, its performance decreases significantly. This suggests that retaining hard-to-learn examples is unfavorable to model performance when

the amount of data is sparse.

### 5.3.2. DATA PRUNING IN DIFFERENT EPOCH

The DAD score quantifies the learning difficulty of the examples; However, is the learning difficulty of the examples fixed? Recent works have shown that there are distinct stages to training in deep neural networks(Jiang et al., 2020; Mangalam & Prabhu, 2019; Faghri et al., 2020; Achille et al., 2017). In this end, we calculate the DAD scores of samples at different training periods, to obverse whether the learning difficulty of the same sample in different periods is consistent. As shown on Fig.6(a), we select 40% of the most hard-to-learn samples from the MSD Pancreas dataset used DAD score computation in different epoch, then we calculate the overlap between the datasets divided from epoch

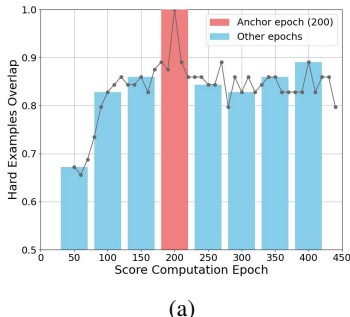
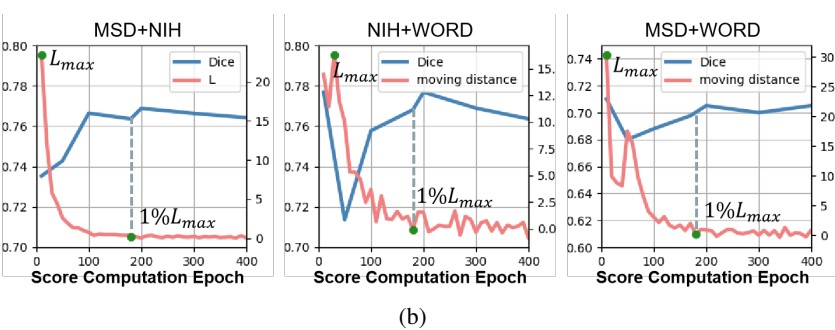

(a)                                    (b)

*Figure 6.* (a) Statistics similarity of subsets (40% hardest to learn of the whole dataset, ranked by DAD score) selected in different score computation epoch w.r.t it selected in anchor epoch. (b) The moving distance curve of data map, and data pruning in different epoch. A good correlation is shown between the two curves.

10 to epoch 400 (red is the anchor epoch, compare with other blue bar). We found the rank of examples learning difficulty are not fixed, there are dynamic change process, and tend to be stable as model training. Therefore, subsets selected in early-stage training and in late-stage training are not identical, for example, 40% of the most hard-to-learn samples select in epoch 10 relative to epoch 300 have only overlap of 0.56.

The reason for this phenomenon is that the random initialization of the network makes it more inclined to learn some samples at the beginning of training, rather than simply dominated by the difficulty of sample learning. For efficiency reasons, its uneconomical to wait a long time for the model to converge before pruning the dataset. Therefore it is important to have a suitable criterion for when to data pruning.

## 6. Discussion

**Task.** Previous method often computes score of the whole image, it does not work on the dense labeling task, especially for small target prospects, where the computed scores are easily affected by the large number of backgrounds and do not pay good attention to the learning difficulty of the target itself. The improved EL2N score (EL2Nx) proves the idea that when we focus only on the prediction accuracy of segmented targets, better results are obtained than focusing on the whole image. And DAD score balances the two (foreground and background) well, with the best performance on almost all pruning scales.

**Selected criteria.** Our method surpass gradient-based methods such as GraNd and VOG. These methods require a large amount of gradient information to be saved, resulting in a huge memory occupation, and they require additional reasoning time to back-propagate the input image. The approximate version of GraNd, EL2N abandons the operation

of gradient calculation and has similar performance with GraNd. Our experiment also compares VOG and EL2N, and no obvious difference is found. The table of memory usage with different method can be found in supplementary material.

**Dataset size.** The latest progress of deep learning often depends on larger models and more data. This will undoubtedly improve the robustness and generalization performance of the model, but there is also a potential problem: unlimited growth of resource consumption. The increase of data volume may also be mixed with a large amount of noise information and redundancy, which is obviously uneconomical. We trimmed the dataset by DAD score and made a preliminary analysis. It was found that the increase in the number of training data does not always mean an increase in performance, but may lead to a decline in the performance of the model.

## 7. Conclusion

In this work, we propose DAD, as an efficient and practical method to rank data according to the learning difficulty of samples, and validate its effectiveness for pruning datasets through a series of experiments. Unlike past methods, DAD score that focuses more on the segmentation target than on the whole image, this is more effective for tasks with dense labels. In addition, we use the DAD score on three different combined datasets to verify that larger data size does not necessarily mean better performance. We believe that the method can be well generalized to other areas, such as small target recognition and active learning, and our future research will focus on these areas.

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
