# OpenReview forum: "Data-Centric Diet: Effective Multi-center Dataset Pruning for Medical Image Segmentation"
_ICML.cc/2023/Workshop/IMLH — IMLH 2023 Poster_

### Official Review · Reviewer_Uwqe · 2023-06-15
**This paper explores how to efficiently train a medical image segmentation network by pruning data from multi-center datasets. The author proposes to use the Dynamic Average Dice (DAD) score to evaluate sample importance and select a subset for training.**

**Rating:** 5
**Confidence:** 4

**Review:**

### Strengths
- The DAD score is effective to assess the importance of samples for medical image segmentation and outperforms the methods proposed for classification task.
- interesting and good experimental results

### Weaknesses
- The motivation for pruning data for multi-center datasets is ambiguous.
- The first contribution claimed in the intro is limited. They combine two datasets to naively train a segmentation model and evaluate the performance on a single dataset. It is a common problem that introducing another center dataset will lead to domain gap. However, they don't explain and analyze the reasons for yielding better or worse models in Fig.1.
- Instance hardness has been widely studied in hard example mining and curriculum learning. Their evaluation mainly depends on the instantaneous or historical training losses with respect to ground truths. Besides, using entropy map to identify the importance of samples is explored for active learning. However, they don't review the related works and compare them.

### Questions:
- The equation(1) is correct?

### Typos issues:
- In line 105, "we we further design"
- Sentence (can use to) in 290 is confusing, hard to understand.
- "Fig.3 presents the instances have the high DAD score." in line 319.
- Fig.6(b) legend for MSD+NIH, the red line L

---

### Official Review · Reviewer_niMG · 2023-06-15
**The paper is well-motivated but still immature**

**Rating:** 6
**Confidence:** 4

**Review:**

The paper defines the moving average of the Dice score as dynamic Dice score and proposes a data pruning method which selects hard-to-learn samples based on the dynamic Dice score.

The strength and weakness of this paper are following:

Strength:

- The paper identifies an important problem for segmentation through the learning dynamics

Weakness:

- The title is about multicenter dataset, but the aspect of multicenter is not fully elaborated in the text. It is unclear how the multi-center dataset matters in the proposed method.

- Writing needs to be improved. There are a lot of notations without definition.

- Missing some related works that leveraged learning dynamics in segmentation [1]

[1] Liu, S., Liu, K., Zhu, W., Shen, Y., & Fernandez-Granda, C. (2021). Adaptive Early-Learning Correction for Segmentation from Noisy Annotations. 2022 IEEE/CVF Conference on Computer Vision and Pattern Recognition (CVPR), 2596-2606.

---

### Official Review · Reviewer_snjD · 2023-06-16
**An interesting exploration of data pruning yet lacking in explanation and motivation**

**Rating:** 4
**Confidence:** 4

**Review:**

This paper introduces a data pruning method for segmentation tasks, utilizing a Dynamic Average Dice score to rank the data samples.

Pros:

1. Easy to follow and understand
2. The authors conduct experiments on multiple datasets

Cons:

1. The writing quality of the paper needs improvement. There are quite a few spelling, grammar problems and unclear notation explanations (e.g., T, t, N, d).
2. The proposed method assumes that samples with lower prediction accuracy are less useful, but this assumption lacks sufficient evidence. It is recommended that the authors explore the differences between hard and easy samples and investigate whether hard samples are noisy data.
3. To strengthen their findings, it is suggested that the authors test their pruning strategy on current advanced methods/networks for dataset segmentation tasks using the same datasets.

---

### Official Review · Reviewer_RWRr · 2023-06-17
**Not good enough**

**Rating:** 4
**Confidence:** 5

**Review:**

Paper type: 8-page long paper.

## Summary

This paper proposes a method to prune the size of datasets for dense labeling problems (e.g. semantic segmentation) without sacrificing model performance. The proposed method utilizes the dynamic average dice (DAD) score, which is calculated over a period of time for each image in the training process, to determine which images to keep. The paper suggests that the time period used for calculating the DAD score should be between the early training stage and the overfitting stage. It also creates a new metric called EL2Nx, which is based on the EL2N metric, to prune the dataset. EL2Nx only focuses on the foreground target, marking it a better metric than EL2N to prune the datasets in the dense labeling problems. The paper shows that the model trained on the pruned dataset based on the DAD score metric achieves competitive performance to the model trained on the full dataset (however, the experiment setup is not clear, See below). It also shows that combining multiple datasets does not necessarily lead to better performance.

## Strength

Although there have been numerous studies on pruning datasets for image classification tasks, there has been little research on pruning datasets for dense labeling problems like semantic segmentation. Therefore, this paper’s idea of using the average dic coefficient, which is a common metric for semantic segmentation, to prune datasets for dense labeling problems is novel. However, I am not fully convinced of the effectiveness of the proposed method, which I will discuss further in the Weakness section.

This paper has conducted studies on three real-world CT datasets and selected two appropriate baseline (VOG and EL2N). Experiments in Figure 4 and Table 1 partially demonstrate the effectiveness of their method, showing that the model trained only with a subset of the dataset can achieve better performance than the model trained with the full dataset.

## Weakness

It is unclear how the multi-center dataset plays a role on the main contribution of this paper given that it is emphasized throughout the paper (e.g. Figure 1, Figure 2, and the title). Is it a necessary element to prune the dataset effectively? If so, why? I think this paper does not answer these questions clearly.

In Figure 1, we observe that combined datasets can both have both positive and negative effect on model performance. For example, the model trained of MSD + WORD perform worse on the MSD test sets, while the model trained NIH + WORD datasets actually performed better on both NIH and WORD test sets. This contradicts with the paper’s claim that “combining multiple training dataset does not necessarily yield better models”. The impact of the data diet on these two kinds of combined datasets remains unclear. The closest investigation into these questions in the paper is Figure 4. However, it is unclear what is the source of the y axis (Pancreas Dice) in each graph. Therefore, I am not sure how useful the proposed method is in the multi-center datasets, which weakens the novelty of this paper.

If a multi-center dataset is not necessary to prune the dataset with DAD score, then I think the study of the proposed method should begin with a single dataset. This leads us to the experiment shown in Table 1. Table 1 shows the different pruning strategies based on the difficulty of the samples. The definition of the sample difficulty is unclear in the paper. What is an easy-to-learn sample? How high is a DAD score considered high? Similar question goes to “ambiguous” and “hard-to-learn” samples. These are not mentioned in the paper. The setting of Table 1 is also unclear, for example, what is the training and testing dataset? These are not mentioned in 5.3.2. What is more, there is no comparison with baselines in this single-dataset setting. All of these weaken the claim that “preserving those ambiguous examples is better than preserving other regions”.

The calculation of DAD depends on some floating parameters \delta_t and the training stage t1, t2. While t2 is defined by the motion distance curve, the value of t1 is not mentioned in the paper. How to select t1, and correspondingly \delta_t? Would their value affect the pruning quality? An ablation study of this would be critical to evaluate the performance of the proposed pruning method.

IOU (intersection over union) is another common metric used to evaluate model performance in semantic segmentation. Since the paper investigates EL2Nx, which is essentially the pixel error of the foreground target, it would be interesting to see how the pruned dataset based on the IOU metric affects the model performance compared to DAD and EL2Nx. I believe this is another reasonable baseline.

The t0 is equation 6 is undefined. Yet it is critical to calculate the moving distance.

Supplementary material is mentioned in line 405 but not provided.

The meaning of the color of dots in Figure 5 is not mentioned.

The writing of this paper could be improved. I would suggest first introducing the experiment setup before discussing the results. Here are some minor writing suggestions:

1. Duplicate “we” in line 106.
2. Duplicate caption letters in line 149.
3. Incorrect caption of “Network” in line 208.
4. Grammar usage, punctuation usage. (e.g. line 375).

## Rating and Justification

The concept of pruning dense labeling datasets in this paper appears novel. However, the experimental setup used to evaluate this concept in a single dataset (Table 1) and combined dataset (Figure 4) is ambiguous. We are also missing a comparison with the baseline in the single dataset setting. As a result, it is difficult to assess the effectiveness of the proposed method. I would advise the author to improve their experimental results and resubmit the paper in the future.

---

### Meta-Review · Area_Chair_3zVD · 2023-06-20

**Recommendation:** Accept (Poster)
**Confidence:** 5

**Metareview:**

This paper received relatively diverse comments from reviewers. The proposed method of using a data pruning approach to address the multi-center data problem is of some interpretability of the workshop topic. However, the paper lacks some fundamental literature studies and comparisons, as reviewers commented. Though there are several major concerns, the writing quality, experiment design and results presented are above average of submission. The author can further improve the quality of the paper by addressing the reviewers' comments.

---

### Decision · Program_Chairs · 2023-06-20

Accept (Poster)